# Expanding Quality by Design Principles to Support 3D Printed Medical Device Development Following the Renewed Regulatory Framework in Europe

**DOI:** 10.3390/biomedicines10112947

**Published:** 2022-11-16

**Authors:** Livia Adalbert, S P Yamini Kanti, Orsolya Jójárt-Laczkovich, Hussein Akel, Ildikó Csóka

**Affiliations:** Institute of Pharmaceutical Technology and Regulatory Affairs, Faculty of Pharmacy, University of Szeged, Eötvös Utca 6, H-6720 Szeged, Hungary

**Keywords:** 3D printing, customized medical device development, new medical device regulation and quality by design

## Abstract

The vast scope of 3D printing has ignited the production of tailored medical device (MD) development and catalyzed a paradigm shift in the health-care industry, particularly following the COVID pandemic. This review aims to provide an update on the current progress and emerging opportunities for additive manufacturing following the introduction of the new medical device regulation (MDR) within the EU. The advent of early-phase implementation of the Quality by Design (QbD) quality management framework in MD development is a focal point. The application of a regulatory supported QbD concept will ensure successful MD development, as well as pointing out the current challenges of 3D bioprinting. Utilizing a QbD scientific and risk-management approach ensures the acceleration of MD development in a more targeted way by building in all stakeholders’ expectations, namely those of the patients, the biomedical industry, and regulatory bodies.

## 1. Introduction

3D printing (3DP), also known as additive manufacturing (AM), has revolutionized medical device (MD) development and its scope of personalized application. Generally, MDs are widely used in the healthcare system in prevention, diagnosis, monitoring, and maintenance of therapy, including curative, supportive, and palliative, as well as in rehabilitation. According to the World Health Organization (WHO), in 30 years almost 35% of the European population will be over 60 years of age, as compared with 25% at present [1]. This will have major consequences on health care systems; therefore, a wider use of innovative MDs will be paramount for the early identification of people’s health issues and their adequate solutions for them. Enabling patients to monitor their health parameters and become more responsive towards their own health will consequently reduce the length of hospitalization. Additionally, in the current COVID stricken healthcare systems, we are witnessing increased demand for medical devices due to the increased prevalence of chronic diseases, and also a demand for portable devices to support a more sustainable personalized homecare setting.

At the same time, the advent of 3DP is opening up a new way to design and develop tailored MD [2]. The increasing medical needs of the aging and vulnerable population to address their individual health issues, combined with increased patient expectations, have catalyzed the incorporation of 3D bioprinting as a new technology for innovative solutions, for bone prostheses, scaffolds with other bioprinted tissues and organs, personalized drug delivery systems, biosensors, new medical devices used during the COVID-pandemic, and other digital devices [2,3,4]. The current paradigm shift also affects biomedical companies and requires them to stay competitive in the constantly changing market, hence it is critical for them to implement steady product development and strategies to customize them for the needs of patients, especially those with chronic disease. In line with European Medicines Agency’s (EMA’s) strategic reflection for 2025, it became paramount to endorse “catalyzing the integration of science and technology in pharmaceutical development” [5] through the innovative use of digital tools and data. These techniques offer a cutting-edge opportunity for the production of complex customized products designed to address the needs of the individual patient [5]. This concept is truly aligned with the vast potential of 3D bioprinting utilized in MD development and production. 

However, MD design poses multiple challenges for design engineering, particularly considering the current change created by the introduction of new medical device regulations (MDR) within the European Union (EU). MD development involves a complex collaboration between regulatory bodies, a highly diverse user base, complex interfacing and vital procedures, and a broad range of interconnected sciences [6]. To address the complexity of the medical and pharmaceutical industry, the US Food and Drug Administration (USFDA) and the EMA continue to work hand in hand to emphasize the importance of adopting a Quality by Design (QbD) quality management framework to facilitate successful production development in a flexible way, integrating the expectations of the patients, as well as the regulatory and biomedical industries [7]. Implementing QbD principles in early phase development would facilitate successful knowledge transfer and ensure a continuous lifecycle product management tool, which would further the acceptability of this profoundly innovative technology of 3DP and ensure its regulatory compliance [8,9]. The introduction of the new MDR prioritizes the safe development of MDs, therefore as adversities will be analyzed, it will support the paradigm shift away from unknown risks towards the future application of 3DP [10].

### 1.1. What Does 3DP or AM Mean?

3DP bridges art and science to print in a new dimension, applying 3D printers to metamorphose 3D computer-aided designs (CAD) into life-changing products, creating more effective and patient-friendly pharmaceutical products and bio-inspired medical devices [11]. 3DP was first developed almost three decades ago and it shook up the entire industrial and scientific fields, providing swift and precise manufacturing of structures and components with high levels of complexity that were not available via conventional methods [12]. It covers a broad range of techniques, such as deposition, binding, or polymerization of materials in successive layers for the manufacture of a variety of drug delivery systems, medical devices, and complex biomedical employments [13,14]. 

In 2010, the American Society for Testing and Materials (ASTM) laid down a set of standards which classifies the range of additive manufacturing processes into seven categories based on the material and technology used [15]. Medical applications of additive manufacturing can be categorized as: (1) medical models; (2) implants; (3) tools, instruments, and parts for medical devices; (4) medical aids, supportive guides, splints, and prostheses; and (5) biomanufacturing [16]. The seven ASTM standardized AM technologies are:

**(A) The Binder jetting** technique uses two materials: a binder and a powder-based material. The building material is present in the powder form and the binder is in liquid form as it acts as an adhesive between the layers of powder. Binder jetting also allows color printing and uses polymers such as acrylonitrile butadiene styrene, polyamide, polycarbonate, metal (stainless steel) and ceramic materials (glass) in this process. This technique of printing is self-supported within the power bed. Typically, binder jetting is not used for tools, instruments, or parts for medical devices [17,18,19,20].

**(B) Directed energy deposition** includes a more complex printing process that is used to repair or add additional material to existing components. This process uses material in either powder or wire form. The machine consists of a nozzle placed on a multi axis arm through which melted material is provided onto the surface and then solidifies. This technology is like the material extrusion process, but the nozzle moves into all directions, as it is not fixed to a specific axis. The typical thickness of a layer falls between 0.25 mm–0.5 mm. The process uses metals as materials such as cobalt chrome and titanium. This technique is utilized rarely and only in implants [21,22].

**(C) Material Extrusion** is the most common technique used for 3D printers and a common extrusion process is fused deposition modelling. This technique prints layer by layer by drawing the material through a nozzle, where it is heated [23,24]. The nozzle moves horizontally, and the platform moves vertically up and down after each layer is formed. The typical thickness of the layer should fall between 0.178 mm–0.356 mm [25]. The material extrusion process is a beneficial process as it can create readily available models with good structural and high-quality properties. This technique’s medical application is broad [19,20,26].

**(D) Material jetting** involves the building of a model or structure, layer by layer. Material is jetted through the nozzle on to the building platform either by using the drop on demand technique or the continuous approach [27,28]. The material is filled in the nozzle which then moves horizontally on the building platform. Later, the layers can be cured by using UV light. The commonly used materials are polymers and waxes because of their viscous nature and ability to form drops. Material jetting is not utilized for implants and biomanufacturing. Polymers used are Polypropylene, HDPE (High density Polyethylene), PS (Polystyrene), PMMA (Polymethylmethacrylate), PC (Polycarbonate), ABS (Acrylonitrile Butadiene Styrene) & HIPS (High Impact Polystyrene) [16,29].

**(E) Power bed fusion** either uses electron beam or laser to melt down and fuse material and powder together. It includes the commonly used techniques such as selective heat sintering, selective laser melting, selective laser sintering, electron beam melting and direct metal laser sintering [30]. All of these processes include the spreading of powder material over previous layers. This can be performed by different mechanisms, such as a roller or a blade. Fresh material is supplied by a hopper or reservoir. This technique’s medical application is broad [19,20,31].

**(F) Sheet lamination** includes two types of techniques: ultrasonic additive manufacturing and laminated object manufacturing. Ultrasonic additive manufacturing involves the use of metal sheets or ribbons such as aluminum, copper, stainless steel and titanium, which are then bound together by using ultrasonic welding. The laminated object manufacturing technique involves a layer-by-layer approach and uses paper as material and adhesive instead of welding. This technology is utilized rarely for medical models or phantoms [16,19].

**(G) VAT Photopolymerization** uses a model that is created layer by layer using vats of liquid photopolymer resin [32]. A UV light is used to cure the resin into the desired model and the platform moves downwards accordingly to build new layers on top of the previous ones [33]. The typical thickness of the layer for the process should be 0.025–0.5 mm. As this technique uses liquid for forming models or structures, there is no structural support provided during the printing phase [34]. Hence, in this case, support structures are needed to be added. After completion of the process, the model must be removed from the resin and the excess resin present in the vat should be drained. This technique’s medical application is broad [19,20,30].

The pillars of the personalized drug delivery approach incorporate the aspects of ‘Design’, ‘Develop’, and ‘Dispense’ that are mandatory for successful individualized therapy. All three aspects involve artwork preparation using computer simulations or CAD of the objects of interest [35]. 3DP-based production can provide personalized therapeutic solutions for individual users, or patients with specific co-morbidities, making 3DP extremely useful in therapies of complex medical profiles, such as epilepsy, Alzheimer’s disease, or cancer, particularly in pediatric and geriatric populations [36,37]. Individualized therapy promotes delivery of the right dose at the right time, maximizing the benefits of drug therapy to achieve the required pharmacokinetic (PK) and pharmacodynamic (PD) responses, while taking into account other parameters, such as genetic makeup, gender, age, and weight to determine the required dose titration and forms [38]. In future clinical pharmacy practice, the pharmacist might use 3DP technology to dispense multiple medications immediately in a tailor-made format following the physician’s prescription [39]. The 3DP concept facilitates the use of alternative therapeutic systems, such as transdermal drug delivery systems that ensure that the active pharmaceutical ingredient (API) is delivered directly into the systemic circulation by skipping the first-pass effect of the liver, achieving a speedier onset of action. A 3DP nasal patch containing polylactic acid combined with salicylic acid has been developed for acne treatment. Moreover, 3DP has been used to prepare transdermal patches with microneedles to deliver biomacromolecules, such as insulin, growth hormone, melanostatin, and erythropoietin, directly into the skin, instead of using needles [14,40]. Coating the microneedles is conventionally challenging but easy with 3DP, where piezoelectric-driven material jetting is applied to specifically coat with the drug solution [41]. A team from Stanford University created a promising 3DP vaccine patch which elicits a great immune response and also allows for self-administration for those who prefer alternative administration routes [42]. 3DP can also be utilized for rectal or vaginal drug delivery systems via suppositories, pessaries, intrauterine devices, and surgical stents, and are also applied to provide drugs for local and systemic therapeutic effect [43]. Finally, using CAD simulations and imaging of a given physiological structure or body cavity can allow for precision dosage forms, achieving individualized and patient-centric drug therapy [44,45]. Table 1 represents the application of 3DP medical devices.

### 1.2. Scope of 3D Bioprinting 

Various types of additive manufacturing techniques have been developed which include using cells and biomaterials for the fabrication of tissues and organs such as inkjet-based 3D bioprinting, laser assisted 3D bioprinting, stereolithographic based 3D bioprinting, and extrusion-based 3D bioprinting [46].

**Inkjet based 3D bioprinting** involves the passing of sequential drops of bioinks on a surface using thermal piezoelectrics or electromagnetic effects. It includes the usage of cells and biomaterials. Cells include neonatal human dermal fibroblasts, dermal microvascular endothelial cells, epidermal keratinocytes, and human chondrocytes. Biomaterials include collagen, fibrinogen, thrombin, poly (ethylene glycol) and dimethacrylate (PEGDMA). This technique can be used for better wound contraction and tissue integration, and comes with some benefits such as its easy availability, inexpensive cost, and that it is a high-speed technique. However, it also lacks precision [47,48,49].

**Extrusion-based 3D bioprinting** involves the passing of bioink through a nozzle under pneumatic or mechanical forces. Cells used include human keratinocytes, fibroblasts, chondrocytes, HUVECs (Human umbilical vein endothelial cells) and MSCs (mesenchymal stem cells). Biomaterials include alginate, hyaluronic acid, fibrinogen, glycerol, collagen, and gelatin. This technique is used for the reconstruction of facial wounds, cartilage or 3DP of endothelialized-myocardium-on-a chip. The benefit of this technique is that a high viscosity bioink can be used and cells can be printed in greater density, but the cell’s structure can be distorted because of the undue stress of density [23,50].

**Laser assisted 3D bioprinting** is used to evaporate bioink and cell suspensions placed at the bottom of a ribbon, which is then moved on to the receiving substrate. The cells used include HUVECs, mouse fibroblasts, human keratinocytes, and MSCs. Biomaterials cover collagen, nanohydroxyapatite and human osseous cell sheets. This technique can be used for soft tissue regeneration and bone and skin grafting. A high degree of precision and resolution can be achieved in this process, but it is an expensive and time-consuming technique [51,52,53].

**Stereolithography based 3D** bioprinting is used to cure photocurable bioink in a layered manner. Cells include the MCF-7 breast cancer cell, HUVECs, C2C12 skeletal muscle cells, osteoblasts, fibroblasts, mesenchymal cells, BrCa (breast cancer) and MSCs. Biomaterials include PEGDA (Poly (ethylene glycol) diacrylate) and GeIMA, GeIMA and nHA (Nano-hydroxyapatite). This technique can be used for preparing a model for post-metastatic breast cancer progression investigation in bone. A high degree of accuracy can be achieved with low printing time, but this technique is highly complicated and also has a lengthy post-processing time [54,55,56].

Generally, native tissues are of more complex structures than engineered constructs, and therefore it is critical to develop functional and biomimetic tissue-like constructs [57]. It is also important to consider the different development stages that the printed tissues often need to undergo, including cell viability, implantation, integration, and remodeling in vivo. Mimicking heterogeneous and complex native tissues have been made possible by bioprinting and by using multicomponent crosslinkable bioinks that can solidify to maintain stable constructs. In principal, shear-thinning biomaterials that are injectable under the application of shear force, show a capacity to quickly self-heal, and are popular as they reform once the shear stress is removed [58]. Challenges related to multicomponent bioinks include the fast degradation properties of hydrogels that have been used for 3D bioprinting of heterogenous and biomimetic structures.

The biomedical applications of 3DP include the creation of artificial structures [59], for example to replace deformed bones in orthopedic [60,61] and dental [62] surgeries, or to form facial reconstruction, artificial ears, noses [63], hearts [64], cornea [65], retina [66], or cardiac pacemakers [67]. CAD programs use CT scans and MRI imaging to develop surface topographic images of complex physiological structures, which then guide 3DP devices to create very precise, artificial structures to replace damaged organs [68,69].

The advantage of 3DP tissues and organs for transplantation will greatly affect transplantation lists, given the shortage of donor organs [70,71]. A primary function of bioprinting is to create a functional physiological microenvironment where the essential printed scaffold is done for subsequent cell seeding or direct cell printing [72,73,74]. Biomaterials such as collagen, hydrogels, and other fabricated materials are applied in bioprinting to directly build 3D tissue-like structures. Moreover, stem cells, cultured on hydrogel substrates, are also applied as bioinks for printing [75,76]. A range of functional characteristics, such as material strength, porosity, and network structure determine the quality of printed organs [77]. The low cost of printing, high accuracy in organ replication, and rapidity of manufacturing, imaging, and visualization are additional advantages of 3D bioprinting over traditional regenerative methods [78]. IJP and material extrusion printing are typical printing methods applied for printing tissues and organs. Using these techniques, bioink or functional material is deposited on a substrate in accordance with a digital image of the required organs or tissues that was produced by CAD simulation software. When printing complex organs, where blood vessels and capillaries are implanted within the structure, a layer-by-layer deposition technique produces the required spaces within the organs [79]. Figure 1 represents the steps of the 3D bioprinting methodology.

A review of the literature indicates that the 3DP methodology has been successfully used to 3DP organs, including whole human organs such as the heart and heart valves from human tissue [80]. A mix of hydrogels originating from fatty acids with human tissues was applied as a bioink to create the organ by IJP [81]. The printing process took 3–4 h to print a heart with basic blood vessels [82]. A successful mitigation of tracheobronchomalacia (TBM) with 3DP personalized medical devices in three pediatric patients grabbed the world’s attention [83]. According to the literature, a multidisciplinary team designed a prototype that allows “radial expansion of the affected airway during the critical growth period, while resisting external compression and intrinsic collapse” [83]. The team reported that the three pediatric patients with severe TBM were successfully treated with the 3DP-ed personalized bioresorbable MD [83].

3DP has also been attempted for creating urinary bladders at Boston Children’s Hospital [84]. Furthermore, 3DP has been considered for the production of human eyes, ears, retina, and cornea in order to restore and replace accidentally damaged structures and birth defective structures [85]. 3DP has been especially beneficial in orthopedic indication as bone prostheses to replace certain parts of bones, joints or discs due to age-related or trauma-related damage, or as a result of birth defects [86,87], as well as in complicated reconstructive dental surgeries [88]. The release of drugs with complex release profiles can also be achieved using 3DP implants [89]. Overall, there is an increasing substantial scientific literature to support the “advantages of digital healthcare procedures with the aid of bioprinters to make reconstruction, transplantation or regeneration of the damaged organs and tissue constructs of the human body possible” [42,90].

It is very common for amputees to wait weeks or months to receive their prosthetics through normal manufacturing routes. However, 3D bioprinting speeds up the whole process and can create functionally identical products more rapidly and inexpensively [91]. The lower price is an advantage for children, as they quickly outgrow their prosthetic limbs. The first use of 3DP prosthetics dates back to 2012 [92]. A novel technique of bone tissue regeneration is to use scaffolds as a temporary structure that induce bone regeneration and that will eventually be absorbed by osteoclasts. The scaffolds support the weight-bearing movements of the patient, while the underlying physiological process of bone tissue regeneration is achieved [93,94]. Selection of suitable materials in bioprinting is critical, as they determine the performance of a particular application. In the selection of bio-materials, the following aspects need to be carefully considered such as printability, biocompatibility, byproducts, kinetics of degradation, mechanical and structural characteristics, and material biomimicry [95].

Furthermore, using 3D medical imaging, virtual surgeries now allow cooperation between the surgeon and the specialized engineering team to study and manipulate within a virtual environment and create the optimal solution for the restoration of the defective part or organ to its physiological function [96]. While a virtual operation strategy can identify hindrance, constraints, and risks of the surgery, it also improves technical performance and confidence during the actual procedure and makes technically impossible surgeries possible [86,97].

**Table 1 biomedicines-10-02947-t001:** Application of 3D printed Medical Devices.

Type of Use	Examples	Advantages	Ref
Personalized therapy	Personalized compounding	Suitable for geriatric use, pediatric use, or for patients with Alzheimer’s disease	[98,99,100]
Wearable devices	3DP-ed electronics for MD	Enables real-time monitoring of chronic medical conditions and transmission of data to a patient’s mobile device	[101,102]
Alternative drug delivery system	Topical masks and wound dressings, transdermal needles	Drug-eluting patches for transdermal application	[100,103]
Personalized MD	Orthodontics, prosthetics, implants vascular stents, orthopedic implants, artificial joints, and heart valves	Drug-eluting implants, resorbable or permanent implants	[73,74,104]
Tissue engineering	Scaffold-based bioprinting for tissues and organs	Potential alternative to transplantation due to lack of human donors	[100,105]
Surgical models	Pre-operative planning and intra-operative guides	Enhances success of surgeries, reduces complications such as blood loss or even patient loss	[106]
Drug Discovery	Drug screening on printed tumor cell lines or other cell lines	Alternative to human or animal models	[107,108]
Microfluidic bioprinting for organ-on-a-chip models	They reflect the structural, microenvironmental and physiological function of human organs	Drug validation testing as an alternative to animal and human models	[109]
Surgical tools	Forceps, hemostats, scalpel and clamps	Low-cost production	[110]

### 1.3. The Advent of Adapting a Quality by Design Strategy

Quality by Design (QbD) was a notion first established by the quality pioneer Joseph M. Juran [111]. Juran believed that the product design should consider product quality, which helps with avoiding the poor initial product design [112]. The US Food and Drug Administration initiated the QbD programs in the pharmaceutical and biotech industries in 2004 to develop carefully designed products, services, and processes, considering all aspects of their lifecycle. Furthermore, QbD had been used and proved to be effective in many other industries for more than 40 years [113].

This concept concentrates on achieving process control via a deep understanding of products and processes applying science, engineering, and quality risk management [114,115]. QbD results in accelerated research timelines and reduced development costs, avoids trial-and-error studies and concentrates on testing methods geared towards product development [112]. The principles of a QbD approach are outlined in the International Council of Harmonization (ICH) guidelines, specifically ICH Q8 (R2) (Pharmaceutical Development), ICH Q9 (Quality Risk Management), and ICH Q10 (Pharmaceutical Quality System) [112,116,117]. The step-by-step cycle diagram in Figure 2 illustrates the QbD model followed for the R&D stage that was previously suggested by Gurba-Bryśkiewicz et al. [118] and then further developed by Martinez-Marquez et al. to facilitate industry translation of custom 3DP bone prostheses [2].

### 1.4. Steps of Quality by Design

The first imperative step of the QbD step-by-step cycle diagram is the definition of the quality target product profile (QTPP), in which the essential parameters of the future product are set from the patient’s and the regulatory point of view, ensuring that the clinical requirements are met [119,120,121]. Beyond users’ needs in relation to product safety and sales, the QTPP should also examine the market success of the relevant device [122,123,124]. The target quality characteristics of customized 3D printed MDs are defined by three sets of quality aspects: the user-based approach, meaning serviceability, aesthetics, and perceived quality; the manufacturing-based approach, focusing on conformability and durability; and then the product-based approach, encompassing performance, features, and reliability. All three quality aspects define our understanding of the market for medical devices [2]. Overall, the expectation of all stakeholders, namely the patients (satisfaction and adherence), users (satisfaction), regulatory bodies and the medical device industry are all incorporated in the definition of QTTP.

Once QTTP is defined, the next step is an effective identification of critical quality attributes (CQAs) based on a scientific and risk management rationale, considering product knowledge and business and regulatory requirements [114]. CQAs can be chemical, mechanical, biological, or microbiological [2], and identifying them is of paramount importance, as they affect the final device quality and performance properties, directly influencing safety and efficacy. Following an extensive review of design and fabrication methods for 3D printed MD, it is recommended to present the research outcome in a fabrication process flow diagram [125]. The next step is the identification of critical process parameters (CPPs) and critical material attributes (CMAs). To collect all of these impacting and relating parameters there are several quality managements tools to be used, for example Ishikawa diagrams, decision tree or Pareto analysis, etc. During the risk identification process, the Ishikawa fishbone diagram visually presents the cause-effect relationship between the critical parameters and the CQAs of the QTTP. The main critical factors are usually grouped into major aspects related to method, machine, materials, and person [126].

Risk assessment ensures a high quality product, identifying and controlling potential risk that is critical during development and manufacturing. Using the appropriate software, the qualitative links form the basis of calculating the severity scores. Results of the assessment are generated and presented in Pareto diagrams, listing numeric data and ranking CQAs and CPPs representing the potential effect on the final product. Assessing the factors beforehand creates a fundamental basis of how these parameters are related. Data must be collected from the literature, and the process requires a comprehensive understanding of regulatory requirements [119,127].

After the risk assessment, categorized risks, associated design, and fabrication processes of the target product need to be evaluated to define the Design Space. Operating within the DS is part of the control strategy, meaning the DS is associated with the control strategy, ensuring that the manufacturing process produces the desired 3D printed MD that meets QTPP. By the implementation of the control strategy, the required device quality can be ensured. 

In essence, QbD saves both time and resources through a better understanding of the CPPs, CMAs, and CQAs, and ultimately develops a robust and reliable production method to optimize product safety, efficacy, and quality from early stage development [128].

### 1.5. New MDR in the EU

#### 1.5.1. Major Improvements in New Regulation

As of 26 May 2021, the 2017/745 MDR officially superseded the 93/42/EC Medical Device Directive (MDD), which had been in effect since 1993 with several necessary updates, due to the emergence of technologies which challenged the previous framework, highlighting gaps and the scarcity of expertise [128].

The new MDR sets high requirements for quality and safety of MDs, and covers two key segments in MD manufacture and distribution within the EU: the manufacturing of a device and the associated software of a device. The new regulation emphasizes market unity and the alignment of requirements. The major improvements brought about by the MDR are the following:

A medical device is now determined as “any instrument, apparatus, appliance, software, implant, reagent, material, or other article intended by the manufacturer to be used for human beings, alone or in combination, for one or more of the following specific medical purposes: diagnosis, prevention, monitoring, prediction, prognosis, treatment, or alleviation of disease” [129] (MDR, Article 2.1) [130].

#### 1.5.2. Reclassification to a Stricter Medical Device Classification

Medical devices are classified based on their risk by applying specific rules (given in MDR Annex VIII1 and in EC guidance 7) that factor in the intended purpose of a device and its inherent risks. Manufacturers of invasive devices designed for implantation, surgeries and other devices described as active, including the software used with such devices, are more strictly classified. These significant changes affected medical devices that previously were class 1 or lower and that now fall into class 2a [131].

MDs in direct contact with the central circulatory system or spinal cord are all defined as high-risk and pass from class II to class III [132]. The new MDR also covers the risk issues of nanomaterials used for medical devices. Those nanomaterials that are in contact with membranes inside the body will fall under the highest risk class, and therefore need to go through the most stringent conformity assessment procedures [133].

#### 1.5.3. Increased Traceability

The MDR introduces a new system of unique device identification (UDI) to enhance the identification and traceability of medical devices. It is applicable to all MDs that enter the market, except for custom-made devices [134]. A new database of medical devices that has been introduced in the European Union, called EUDAMED, provides all the information about the devices and is accessible to all relevant stakeholders, users and regulators [135]. With the introduction of implant cards, patients are allowed to identify the summary of a device’s safety and clinical performance in EUDAMED. Overall, the purpose of the new system is to strengthen market surveillance and transparency in the medical device field [136].

#### 1.5.4. Heightened Attention on the Quality Management System

The quality management system now also incorporates the procedure for clinical evaluation and maintenance of a post-market surveillance (PMS) system, as well as post-market clinical follow-up for every product. This encompasses proactive performance monitoring of the recertification device, annual safety updates for high-risk class devices, and the prompt recording of incidents [137,138].

#### 1.5.5. Tighter Clinical Evaluation Requirements

Clinical evaluation is determined as “a systematic and planned process to continuously generate, collect, analyze, and assess the clinical data pertaining to a device in order to verify the safety and performance, including clinical benefits, of the device” (MDR Article 2.44) [139]. This requires the collection of clinical data available in the literature and the organization of necessary clinical trials. All implantable medical devices and class III medical devices must now undergo clinical trials, with only a few exceptions [132]. For all Class III and IIb devices intended to manage a drug (in or out of the body), the manufacturer may consult with a group of EU experts in order to gain their opinions on the clinical development plan [140].

#### 1.5.6. Supervision of Notified Bodies

The MDR contains more requirements regarding the designation of notified bodies that are now supervised by national competent authorities and the European Commission [141].

#### 1.5.7. Introduction of an Independent Expert Panel

In classes IIa, IIb and III, the notified body now needs to go through conformity assessments of all devices produced for CE marking. As per Article 54 of the regulation, certain Class IIb devices may undergo a non-obligatory consultation procedure by an independent expert panel, while such consultation is obligatory for class III devices intended for implantation. The consultation is based on a clinical evaluation analysis report from a notified body [142].

#### 1.5.8. Flexibility to Allow Innovation in the Renewed Regulatory Framework within the EU

To keep up with future progress, the new regulations contain many provisions to increase security and regulatory certainty, such as harmonized rules on drug-device combination products, tissue engineering, nanoscience, personalized medicine, substance-based devices, and genetic tests. The provisions also take into account the latest developments in the sector related to medical software, apps, and cybersecurity [116].

To support innovation, the new regulatory system flexibly allows special devices to be available under certain circumstances. In other words, such devices are exempt from formal clinical evaluation before use, and they are not CE-marked. If they meet certain conditions, then this covers custom-made devices (MDR Article 21 and Annex XIII), devices manufactured and used within the same health institution (Article 5.5), and devices authorized by competent authorities in the interests of patient safety or public health (Article 59). “A health institution that manufactures a device under Article 5.5 must publicly declare that the device meets general safety and performance requirements” (MDR Article 5.5.e.iii) [143].

Investigational devices that are undergoing clinical evaluation are also not CE-marked (MDR Article 21). Investigators may seek single assessments from several EU regulatory agencies (MDR Article 78). It is then mandatory that comprehensive results of all investigations of new devices are published in the scientific literature. In the case of a genuine, unmet, clinical need in a life-threatening circumstance for an individual patient, physicians can apply to their national regulatory agency for approval to use a device that is not CE-marked. However, regulators have stipulated that experience gained from such ‘compassionate use’ cases cannot be accepted as sufficient clinical evidence for conformity [144]. Overall, the customizability and unique build processes of 3D printed medical devices constitute challenges for regulatory requirements related to quality assurance in manufacturing [145]. Therefore, to prevent unevenness in end-product quality, a QbD approach is advised to be used with an in-depth understanding of critical material attributes, critical process parameters, and in-process analysis, in addition to finalized, product testing parameters for product quality consistency [146,147]. Working within a QbD principles- driven framework, which is supported by the regulatory bodies, is likely to provide leading-edge solutions for quality issues during the developmental phase and could facilitate lifecycle management tools for custom-made devices [148]. Figure 3 summarizes the merits of adopting 3DP in developing an MD.

### 1.6. Barriers to 3D Printed Medical Device Innovation

In the coming years, the technologies and materials for 3DP are expected to grow exponentially, but the applications in the healthcare system may not grow proportionally. The focus on the use of 3DP will likely continue in those clinical applications that are currently the readiest and most tested in adopting this technology [149].

The barriers to the innovation of 3D printed MDs are mainly regulatory issues, intellectual property rights (IPR), patient issues and publication, technology and material failures, ethical considerations, security, medical practice patterns, reimbursement and pricing and subsequently the issue of market size and penetration, meaning the application will most likely be used in developed countries [150], as presented in Figure 4.

Additionally, safety presents a critical regulatory hurdle in the field of 3DP. Especially with regard to the aspect of bio-printing, safety issues focus on the risks linked with undertaking medical procedures outside professional medical settings. A European analysis of 3D bioprinting for medical and enhancement purposes indicates that "the side effects of bio-printing have barely been assessed questions about biomaterial degradation, tissue integration, biocompatibility, and continuous tissue synthesis during material degradation". The printing materials and process also present safety concerns. The application of novel polymers, sometimes incorporated with nanoparticles, may have long term risks for implants, making post-marketing surveillance and registries vitally important [151].

Standardization of the starting materials and analysis techniques for 3DP objects have been flagged as a priority, as has the use of an international database of key performance metrics, which will simplify quality assessment and facilitate the repeatability and safety of the final products [152].

In conclusion, the challenge to large-scale adoption of 3DP is a combination of “high potential/high barriers”. The main obstacles to bringing 3DP into the mainstream healthcare system are regulatory, technological, and technology-related investment challenges [77,153,154]. However, once more experience has been gained with 3DP MDs, then a turning point will be reached, and following the FDA’s approach, the European Union will move in the same direction. First, however, the main challenges need to be overcome before we can fully harness the personalized capacities of this technology, in the form of implantable devices or new delivery systems available directly in hospitals or in clinical pharmacies [39].

### 1.7. Emerging Technologies and Devices Facilitated by the COVID-Pandemic

According to a Eurohealth release, “prior to the COVID-19 pandemic, also known as the coronavirus pandemic 2019, there was much unrealized potential in the use of digital tools across Europe”. The pandemic facilitated and sped up the acceptance of the use of digital health technologies, as many digital health tools became an imminent necessity to support communication, information, surveillance, and monitoring in addition to the rollout of vaccination programs [127,155,156,157,158,159,160].

Beyond the vast palette of quickly developed COVID-related devices, the use of digital health technologies was accelerated with electronic-diaries, remote patient monitoring, quick diagnostic kits and wearable devices, such as “accelerometers to track activity, glucometers to track blood glucose levels and devices to monitor heart rates” [161,162]. Telemedicine ™ is becoming increasingly popular in many medical fields including “Neurology, Ophthalmology, Psychiatry, Dermatology, Pediatrics and Allergy” [163,164,165], especially in the absence of in-person visits. TM overcomes two main obstacles that patients encounter when seeking health care: distance and time. Remote patient monitoring encompasses the use of devices, smartphones and applications that can directly report objective information to the monitor, which excludes data distortion due to patient bias. The availability of the large data set gained from digital monitoring allows further analysis and thus has the potential to finally realize personalized treatment, which supports the current paradigm shift [166,167]. The use of TM and other medical devices, particularly combined with information technologies, have the potential to bring about a transformational change in health care by altering the interaction process between patient and provider [168]. This shift is being supported by the current concept in the EU. According to Eurohealth it is now emphasized t“at “European Union funding and initiatives such as the European Health Data Space will support progress in this ”rea” [127].

Unlike the current system, the use of TM, especially when merged with information technologies such as electronic health records and data, has the potential to transform the way health-care is managed by creating new patient and provider interactions. The nine stages of **transformational change** described by Tipton are the following: “1. Status Quo, 2. Denial, 3. Righteous resistance, 4. Pleading, 5. Despair or skepticism, 6. Tolerance, 7. Acceptance, 8. Agreement, and 9. Advocacy” [157].

Currently, a few leading, developed countries are moving beyond tolerance towards acceptance by most health care organizations, the seventh step out of the nine steps of transformational change as defined by Tipton. According to Tripton’s theory, ‘Once the technology has passed the tolerance stage, it is difficult to return to the old way of care, which was restricted to episodic in-person visits’. Therefore, a shift is anticipated ‘toward agreement and eventually we expect advocacy to become widespread’, at least in developed countries [166]. Tripton’s theory is most likely to be applicable for patients who are opting for the application of new generation of 3D printed digital devices.

The best example of the application of smart 3D printed medical devices is the nano-structure cellulose base-3D printed smart dressing which allows healing and wound monitoring. A tailored 3D printed cast that contains a low intensity pulsed ultrasound system heals damaged bone 38% faster than achieved by the conventional way. Tailor-made heart sensors are life-saving sensors for which 3D printed technology is used to facilitate the stretchy sensor to be built around the patient’s heart [42]. These are just a few examples for the most groundbreaking 3D printed medical devices that have been developed in recent years. 

## 2. Conclusions

Overall, MD innovation has brought enormous benefits to patients, especially in the developed world. 3DP or additive manufacturing swiftly percolated to MD development, harnessing it to create customized devices with unique compositions and structures, targeting unmet clinical needs in the health care system. Powerful capabilities of 3DP introduced new challenges such as patent issues, ethical consideration, regulatory incoherence across the world, hurdles with maturing of multicomponent bioinks in addition to scaling up and standardization of bioprinting processes. Beside these challenges, our review focused on the following findings in relation to 3DP advancement in MD development.
-Quality and safety aspects have been partly remedied by the MDR introduced in the EU in May 2021. The new MDR allows application of this technology for unmet clinical needs, which serves as a first step in the evolution and wider use of this profoundly innovative, customized production method. Regulations are harmonized within the EU that may facilitate the wider use of 3D implants in clinics.-The current standardization methods used for traditional production are not appropriate for 3DP technology. Additive manufacturing incorporates new technologies and continuously emerging new biomaterials in the biomedical field. Therefore, 3DP technology still remains to complement rather than replace traditional manufacturing techniques in the near future. -The design and fabrication of customized implants requires multiple steps that might lead to imperceptible errors, affecting the final product and, consequently, patient safety. Therefore, triumphant knowledge transfers of this new design and manufacturing method in the industry requires more integrative technology transfer, which is concurrent with multi-disciplinary cooperation. -The Quality by Design quality management framework offers an integrative tool to build quality and safety into the product development processes, facilitating the incorporation of changes, iterations and improvements. Based on a risk assessment evaluation, it allows some modification, yet following the appropriate steps in the required roadmap, it ensures a target product with the associated quality requirements. QbD has already preferably and successfully been used in the pharmaceutical industry, and it has the proven capability to successfully foster the wider integration and acceptance of 3DP in MD development and manufacturing.

## Figures and Tables

**Figure 1 biomedicines-10-02947-f001:**
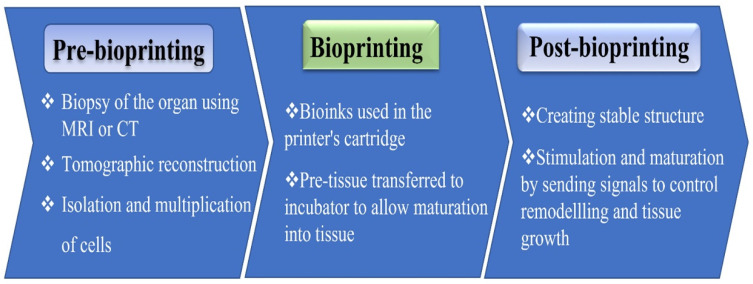
Vital steps of bioprinting: From pre- to post-bioprinting.

**Figure 2 biomedicines-10-02947-f002:**
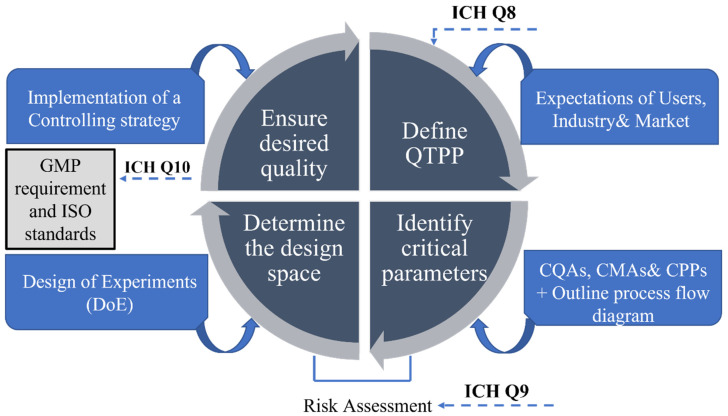
QbD quality management tool showing the principal steps. Abbreviations: ICH: International Council for Harmonization, QTPP: Quality target product profile, CQAs: Critical quality attributes, Critical materials attributes, Critical process parameters, GMP: Good manufacturing process.

**Figure 3 biomedicines-10-02947-f003:**
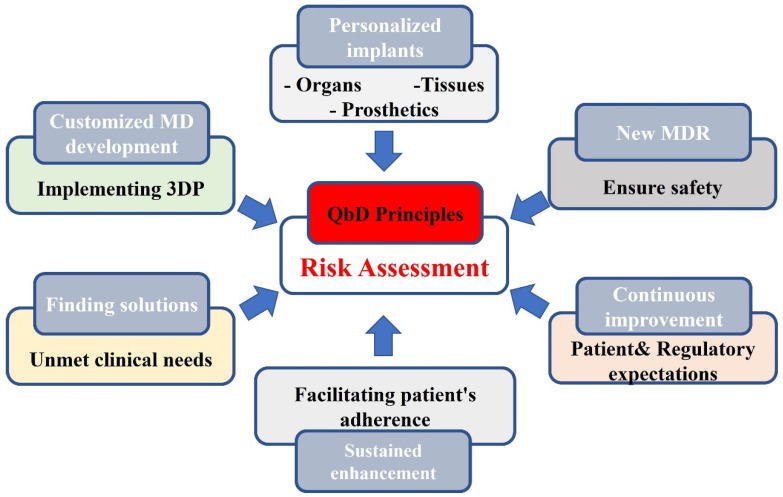
Fostering 3D printed MD development via a QbD quality management framework. Abbreviations: QbD quality by design, 3DP 3D printing, MD: Medical devices MDR Medical Device Regulation.

**Figure 4 biomedicines-10-02947-f004:**
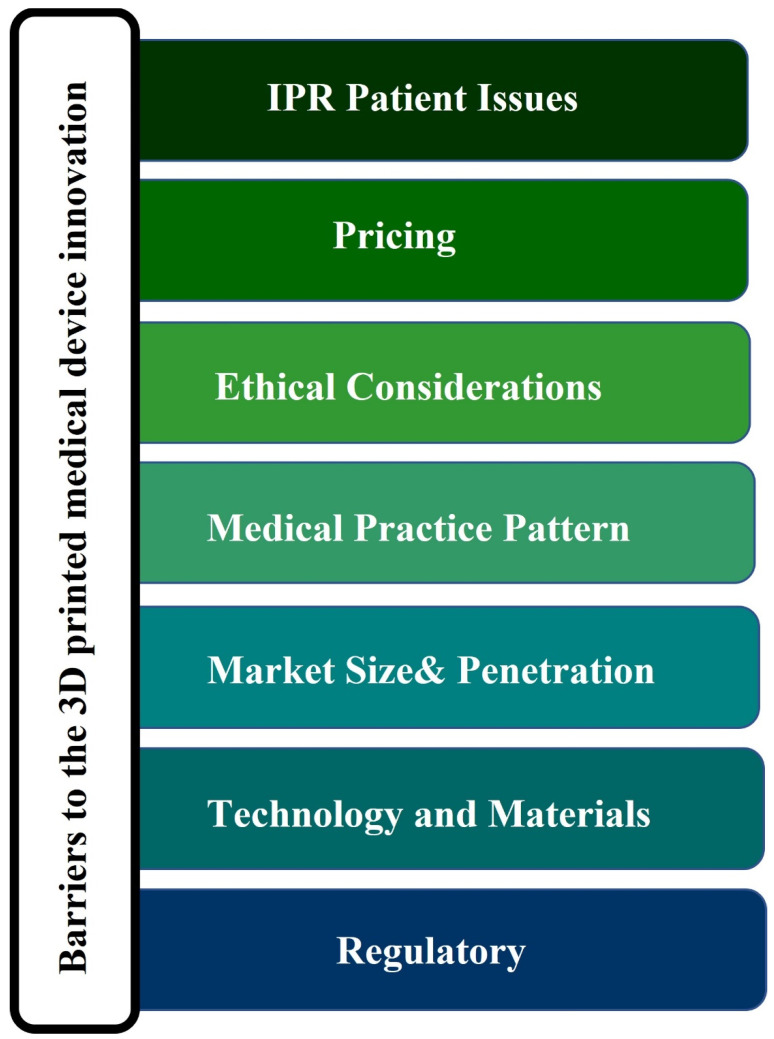
Barriers to the innovation of 3D printed medical devices. Abbreviations: IPR: intellectual property rights.

## Data Availability

Not applicable.

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
