# Peer review of "Expanding Quality by Design Principles to Support 3D Printed Medical Device Development Following the Renewed Regulatory Framework in Europe"

_biomedicines, 2022, doi:10.3390/biomedicines10112947_

Round 1

Reviewer 1 Report

1)     Right at the abstract, the paper uses both 3D Printing and Additive Manufacturing. Indicate that they are the same thing in reality. And, they are used for the same meaning.

2)     In the abstract, ‘We’ is used several times. Make the sentences ‘Passive’.

3)     Once define Additive Manufacturing as AM and 3D Printing as 3DP. And just use AM and 3DP later on.

4)     Re-write the following sentence with the use of seven ASTM standardized AM technologies with a reference.

A few of the most commonly used methods encompass binder jet printing (BJP), fused deposition modeling (FDM), semi-solid extrusion (SSE), selective laser sintering (SLS), Inkjet printing (IJP) and stereolithography (SLA) [13, 16, 17]

5)     For table 1, add another column and provide the key references in that column.

6)     QbD has been defined again and again. Stop doing this kind of repetition.

7)     Design of Experiments could be labeled as DoE.

8)     For Figure 4, use the same style of font. Do not change. Consistency is important.

9)     Define COVID and indicate its impact to this paper. Add a reference.

10)  Rewrite the Conclusions and summarize your key findings as bullets. Your conclusion is generic.

11)  There is no reference from the Biomedicines Journal. Add a few papers from Biomedicines.

12)  In Graphical Abstract, write out MD.

Author Response

                                                                             09/November/2022

Dear Reviewer,

Thank you for sending the reviewer’s comments on our manuscript: “Expanding Quality by Design principles to support 3D printed medical device development following the renewed regulatory framework in Europe.

We are pleased that the reviewers find our review article fairly comprehensive covering a wide range of recent references on the topic. The reviewers requested some editing work and raised a list of valid points that significantly helped in improving the revised manuscript. All corrections appear in track changes and added sections are highlighted with yellow in the manuscript.

All authors have seen and agreed with the contents of the manuscript, and none has any financial interest. The submission is not under review elsewhere.

We hope that you now find the improved manuscript suitable for publication in Biomedicines and look forward to hearing from you.

Yours sincerely,

Livia Adalbert

Detailed response to reviewer

Reviewer #1:

We thank the reviewer for the constructive comments and suggestions. We feel that the manuscript has been considerably improved through the suggested changes.

Reviewer:

1)     Right at the abstract, the paper uses both 3D Printing and Additive Manufacturing. Indicate that they are the same thing in reality. And, they are used for the same meaning.

Author reply:

It has been corrected.

3D printing (3DP) also known as additive manufacturing (AM) has revolutionized medical device (MD) development and its scope of personalized application.

Reviewer:

2)     In the abstract, ‘We’ is used several times. Make the sentences ‘Passive’.

Author reply:

It has been corrected.

This review aims to provide an update on the current progress and emerging opportunities for additive manufacturing following the introduction of the new medical device regulation (MDR) within the EU. The advent of early-phase implementation of the Quality by Design (QbD) quality management framework in MD development is a focal point. The application of a regulatory supported QbD concept will ensure successful MD development, as well as pointing out the current challenges of 3D bioprinting.

Reviewer:

3)     Once define Additive Manufacturing as AM and 3D Printing as 3DP. And just use AM and 3DP later on.

Author reply:

It has been corrected throughout the manuscript.

Reviewer:

4)     Re-write the following sentence with the use of seven ASTM standardized AM technologies with a reference.

A few of the most commonly used methods encompass binder jet printing (BJP), fused deposition modeling (FDM), semi-solid extrusion (SSE), selective laser sintering (SLS), Inkjet printing (IJP) and stereolithography (SLA) [13, 16, 17]

Author reply:

It has been corrected. Additional information provided to include the seven ASTM standardized AM technologies with a reference.

In 2010, the American Society for Testing and Materials (ASTM) laid down a set of standards which classifies the range of additive manufacturing processes into 7 categories based on the material and technology used [16]. Medical applications of additive manufacturing can be categorized as: 1) Medical models; 2) Implants; 3) Tools, instruments, and parts for medical devices; 4) Medical aids, supportive guides, splints, and prostheses; 5) Biomanufacturing [17].

  1. A) Binder jetting technique uses, two materials: a binder and a powder-based material. The building material is present in the powder form and the binder is in liquid form as it acts as an adhesive between the layers of powder. Binder jetting also allows color printing and uses polymers such as acrylonitrile butadiene styrene, polyamide, polycarbonate, metal (stainless steel) and ceramic materials (glass) in this process. This technique of printing is self-supported within the power bed. Typically, binder jetting is not used for tools, instruments, and parts for medical devices [18-21].
  2. B) Directed energy deposition includes a more complex printing process that is used to repair or add additional material to existing components. This process uses material in either powder or wire form. The machine consists of a nozzle placed on a multi axis arm through which melted material provided onto the surface and then it solidifies. This technology is like material extrusion process, but the nozzle moves into all the directions as it is not fixed to a specific axis. Typical thickness of layer falls between 0.25mm- 0.5mm. The process uses metals as materials such as cobalt chrome and titanium. This technique is utilized rarely only in implants [22, 23]
  3. C) Material Extrusion is the most common technique used for 3 D printers and a common extrusion process is fused deposition modelling. This technique prints layer by layer by drawing the material through nozzle, where it is heated [24, 25]. The nozzle moves horizontally, and the platform moves vertically up and down after each layer is formed. The typical thickness of layer should fall between 0.178mm-0.356 mm [26]. The material extrusion process is beneficial process as it can create readily available models with good structural and high-quality properties. This technique’s medical application is broad [20, 21, 27]
  4. D) Material jetting involves building of model or structure, layer by layer. Material is jetted through the nozzle on to the building platform either by using drop on demand technique or continuous approach[28, 29]. The material is filled in the nozzle which then moves horizontally on the building platform. Later, the layers can be cured by using UV light. The commonly used materials are polymers and waxes because of their viscous nature and ability to form drops. Material jetting is not utilized for implants and biomanufacturing. Polymers used are Polypropylene, HDPE (High density Polyethylene), PS (Polystyrene), PMMA (Polymethylmethacrylate), PC (Polycarbonate), ABS (Acrylonitrile Butadiene Styrene) & HIPS (High Impact Polystyrene) [17, 30]
  5. E) Power bed fusion either uses electron beam or laser to melt down and fuse material and powder together. It includes the commonly used techniques such as selective heat sintering, selective laser melting, selective laser sintering, electron beam melting and direct metal laser sintering [31]. All these processes include spreading of powder material over previous layers. This can be performed by different mechanisms, such as a roller or a blade. Fresh material is supplied by a hopper or reservoir. This technique’s medical application is broad [20, 21, 32]
  6. F) Sheet lamination includes 2 types of techniques: ultrasonic additive manufacturing and laminated object manufacturing. Ultrasonic additive manufacturing involves the use of metal sheets or ribbons such as aluminum, copper, stainless steel and titanium, which are then bound together by using ultrasonic welding. Laminated object manufacturing technique involves a layer-by-layer approach and uses paper as material and adhesive instead of welding. This technology is utilized rarely for medical models or phantoms [17, 20]
  7. G) VAT Photopolymerization uses a model that is created layer by layer using vat of liquid photopolymer resin [33]. An UV light is used to cure the resin into the desired model and the platform moves downwards accordingly to build new layers on the top of previous ones [34]. The typical thickness of layer for the process should be 0.025-0.5 mm. As this technique uses liquid for forming models or structures, there is no structural support provided during the printing phase [35]. Hence, in this case, support structures are needed to be added. After completion of the process, the model must be removed from the resin and the excess resin present in vat should be drained. This technique’s medical application is broad [20, 21, 31].

Reviewer:

5)     For table 1, add another column and provide the key references in that column.

Author reply:

Table 1 has been corrected. Additional column with key references have been provided.

Reviewer:

6)     QbD has been defined again and again. Stop doing this kind of repetition.

Author reply:

It has been corrected. repetitive sentences have been taken out.

1.3. The advent of adapting a Quality by Design Strategy

Quality by Design (QbD) was a notion first established by the quality pioneer Joseph M. Juran [114]. Juran believed that the product design should consider product quality, which helps avoiding the poor initial product design [115]. The US Food and Drug Administration initiated the QbD programs in the pharmaceutical and biotech industries in 2004 to develop carefully designed products, services, and processes, considering all aspects of their lifecycle. Furthermore, QbD had been used and proved to be effective in many other industries for more than 40 years [116].

This concept concentrates on achieving process control via a deep understanding of products and processes applying science, engineering, and quality risk management [117, 118]. QbD results in accelerated research timelines and reduced development costs, avoids trial-and-error studies and concentrates on testing methods geared towards product development [115]. The principles of a QbD approach are outlined in the International Council of Harmonization (ICH) guidelines, specifically ICH Q8 (R2) (Pharmaceutical Development), ICH Q9 (Quality Risk Management), and ICH Q10 (Pharmaceutical Quality System) [115, 119, 120].The step-by-step cycle diagram in Figure 2 illustrates the QbD model followed for R&D stage that was previously suggested by Csóka et al. [121] and then further developed by Martinez-Marquez et al to facilitate industry translation of custom 3D printed bone prostheses [2].

1.4. Steps of Quality by Design

I.

The first imperative step of the QbD step-by-step cycle diagram is the definition of the QTTP, in which the essential parameters of the future product are set from the patient’s and regulatory point of view ensuring to meet the clinical requirements [126]. Beyond users’ needs in relation to product safety and sales, the QTPP should also examine the market success of the concerning device [127]. The target quality characteristics of customized 3D printed MDs are defined by three sets of quality: the user-based approach, meaning serviceability, aesthetics, and perceived quality, the manufacturing-based approach, focusing on conformability and durability, and then the product-based approach, encompassing performance, features, and reliability. All three quality aspects define our understanding of the market for medical devices [2]. Overall, expectation of all stakeholders, namely the patients (satisfaction and adherence), users (satisfaction), regulatory bodies and medical device industry are all incorporated in the definition of QTTP.

Once QTTP is defined, then the next step is an effective identification of CQAs based on a scientific and risk management rationale, considering product knowledge, and business and regulatory requirements [117]. CQAs can be chemical, mechanical, biological, or microbiological [2] and identifying them is of paramount importance as they impact the most the final device quality, and performance properties, directly influencing safety and efficacy. Following an extensive review of design and fabrication methods for 3D printed MD, it is recommended to present the research outcome in a fabrication process flow diagram [128]. The next step is the identification of critical process parameters (CPPs) and critical material attributes (CMAs). To collect all these impacting and relating parameters there are several quality managements tools to be used, for example Ishikawa diagrams, decision tree or Pareto analysis etc. During the risk identification process Ishikawa fishbone diagram visually presents the cause-effect relationship between the critical parameters and the CQAs of the QTTP. The main critical factors are usually grouped into major aspects related to method, machine, materials, and person [129].

Risk assessment ensures high quality product, identifying and controlling potential risk that are critical during development and manufacturing. Using the appropriate software, the qualitative links form the basis of calculating the severity scores. Results of the assessment generated and presented in Pareto diagrams, listing numeric data and ranking CQAs and CPPs representing the potential effect on the final product. Assessing the factors beforehand have a fundamental basis of how these parameters are related. Data must be collected from literature and the process requires a comprehensive understanding of regulatory requirements [122, 130].

After the risk assessment, categorized risks, associated design, and fabrication processes of the target product need to be evaluated to define the Design Space. Operating within the DS is part of the control strategy, meaning the DS associated with the control strategy ensures that the manufacturing process produces the desired 3D printed MD that meets QTPP. By the implementation of control strategy, the required device quality can be ensured.

FIn essence, QbD saves both time and resources through a better understanding of the CPPs, CMAs, and CQAs, and ultimately develops a robust and reliable production method to optimize product safety, efficacy, and quality from early stage development. [131].

Reviewer:

7)     Design of Experiments could be labeled as DoE.

Author reply:

It has been corrected. DoE is included.

Reviewer:

8)     For Figure 4, use the same style of font. Do not change. Consistency is important.

Author reply:

It has been corrected to use the same style of font.

Reviewer:

9)     Define COVID and indicate its impact to this paper. Add a reference.

Author reply:

It has been corrected. Additional information provided with references.

1.7. Emerging technologies and devices facilitated by the COVID-pandemic

According to Eurohealth release ‘prior to COVID-19 pandemic, also known as the coronavirus pandemic 2019, there was much unrealized potential in the use of digital tools across Europe’. The pandemic facilitated and sped up the acceptance of the use of digital health technologies as many digital health tools became an imminent necessity to support communication, information, surveillance, and monitoring in addition to rollout of vaccination programs [130, 160-163]. [164].

Beyond the vast palette of quickly developed COVID-related devices, the use of digital health technologies was accelerated with electronic-diaries, remote patient monitoring, quick diagnostic kits and wearable devices, such as “accelerometers to track activity, glucometers to track blood glucose levels and devices to monitor heart rates” [166, 167]. Telemedicine (TM) is becoming increasingly popular in many medical fields including “Neurology, Ophthalmology, Psychiatry, Dermatology, Pediatrics and Allergy” [168-170], especially in the absence of in-person visits. TM overcomes two main obstacles that patients encounter when seeking health care: distance and time. Remote patient monitoring encompasses the use of devices, smartphones and applications that can directly report objective information to the monitor, which excludes data distortion due to patient bias. The availability of the large data set gained from digital monitoring allows further analysis and thus has the potential to finally realize personalized treatment, which supports the current paradigm shift [171, 172]f. The use of TM and other medical devices, particularly combined with information technologies, have the potential to bring about a transformational change in health care by altering the interaction process between patient and provider [173]. This shift is being supported by the current concept in the EU. According to Eurohealth it is highlighted now that ‘European Union funding and initiatives such as the European Health Data Space will support progress in this area’.  (Williams GA: COVID)

Unlike the current system, the use of TM, especially when merged with information technologies such as electronic health records and data, has the potential to transform the way health-care is managed by creating new patient and provider interactions. The 9 stages of transformational change described by Tipton are the following: “1. Status Quo, 2. Denial, 3. Righteous resistance, 4. Pleading, 5. Despair or skepticism, 6. Tolerance, 7. Acceptance, 8. Agreement, and 9. Advocacy” [174].

Currently, a few leading, developed countries are moving beyond tolerance towards acceptance by most health care organizations, the 7th step out of the 9 steps of transformational change as defined by Tipton. According to Tripton’s theory ‘Once the technology has passed the tolerance stage, it is difficult to return to the old way of care, which was restricted to episodic in-person visits. Therefore, a shift is anticipated ‘toward agreement and eventually we expect advocacy to become widespread’, at least in developed countries [171]. Tripton’s theory is most likely to be applicable for patients who are opting for the application of new generation of 3D printed digital devices.

The best example of application of smart 3D printed medical devices is the nano-structure cellulose base-3D printed smart dressing which allows healing and wound monitoring. Tailored 3D printed cast that contains low intensity pulsed ultrasound system heals damaged bone 38% faster. Tailor-made heart sensors are life-saving sensors for which 3D printed technology is used to facilitate the stretchy sensor to be built around the patient’s heart [45]. These are just few examples for the most groundbreaking 3D printed medical devices that have been developed in recent years.

Reviewer:

10)  Rewrite the Conclusions and summarize your key findings as bullets. Your conclusion is generic.

Author reply:

Conclusion has been modified and highlights have been listed in points.

  1. Conclusion

Overall, MD innovation has brought enormous benefits to patients, especially in the developed world. 3DP or additive manufacturing swiftly percolated to MD development, harnessing it to create customized devices with unique compositions and structures, targeting unmet clinical needs in health care system. Powerful capabilities of 3DP introduced new challenges such as patent issues, ethical consideration, regulatory incoherence across the world, hurdles with maturing of multicomponent bioinks in addition to scaling up and standardization of bioprinting processes. Beside these challenges, our review focused on the following findings in relation to 3DP advancement in MD development.

- Quality and safety aspects have been partly remedied by the MDR introduced in the EU in May 2021. The new MDR allows application of this technology for unmet clinical needs, which serves as a first step in the evolution and wider use of this profoundly innovative, customized production method. Regulations are harmonized within the EU that may facilitate wider use of 3D implants in clinics.

The current standardization methods used for traditional production are not appropriate for 3DP technology. Additive manufacturing incorporates new technologies and continuously emerging new biomaterials in the biomedical field. Therefore, 3DP technology still remains to complement rather than replace traditional manufacturing techniques in the near future.

-Design and fabrication of customized implants need multiple steps that might lead to imperceptible errors, impacting the final product and consequently, patient safety. Therefore, triumphant knowledge transfers of this new design and manufacturing method in the industry requisites more integrative technology transfer, concurrent with multi-disciplinary cooperation.

-Quality by Design quality management framework offers an integrative tool  to build quality and safety into the product development processes, facilitating the incorporation of changes, iterations and improvement. Based on a risk assessment evaluation, it allows some modification, yet following the appropriate steps in the required roadmap, it ensures a target product with the associated quality requirements. QbD has already preferably and successfully been used in the pharmaceutical industry, it has the proven capability to successfully foster wider integration and acceptance of 3DP in MD development and manufacturing.

Reviewer:

11)  There is no reference from the Biomedicines Journal. Add a few papers from Biomedicines.

Author reply:

References from Biomedicines hav been included.

Reviewer:

12)  In Graphical Abstract, write out MD.

Author reply:

Graphical abstract has been corrected to include Medical Device.

Reviewer 2 Report

Overall, the paper is well-written. However, the authors need to make some minor changes such as using the standardized ASTM terms for different 3D printing techniques and elaborate more on bioprinting.

11.      In Section 1.1, the authors mentioned the different commonly used 3D printing techniques. The authors should use the ASTM classification for the 3D printing techniques and provide some relevant references for some of the mentioned techniques.

a.      Binder Jetting (binder jet printing)

b.      Directed-Energy Deposition

c.      Material Extrusion (fused deposition modelling, semi-solid extrusion)

d.      Material Jetting (inkjet printing)

e.      Powder-Bed Fusion (selective laser sintering)

f.       Sheet Lamination

g.      Vat polymerization (stereolithography)

22.      In the 2nd part of Section 1.2, the term “bioprinting” should be used when cells are involved during the printing process. The authors can refer to some of the highly cited papers in the field of bioprinting.

a.      "Print me an organ! Why we are not there yet." Progress in Polymer Science 97 (2019): 101145.

b.      "Bioinks and bioprinting technologies to make heterogeneous and biomimetic tissue constructs." Materials Today Bio 1 (2019): 100008.

33.  The authors should also discuss what are some of the bioprinting techniques that are suitable for cell printing? The authors can refer to some recent papers below.

a.      Material extrusion

                                                    i.     "Extrusion bioprinting of soft materials: An emerging technique for biological model fabrication." Applied Physics Reviews 6, no. 1 (2019): 011310.

                                                   ii.     "Layer-by-layer ultraviolet assisted extrusion-based (UAE) bioprinting of hydrogel constructs with high aspect ratio for soft tissue engineering applications." PLoS One 14, no. 6 (2019): e0216776.

b.      Material jetting

                                                    i.     "Inkjet bioprinting of biomaterials." Chemical Reviews 120, no. 19 (2020): 10793-10833.

                                                   ii.     "Controlling Droplet Impact Velocity and Droplet Volume: Key Factors to Achieving High Cell Viability in Sub-Nanoliter Droplet-based Bioprinting." International Journal of Bioprinting 8, no. 1 (2022). 424

c.      Vat polymerization

                                                    i.     "Vat polymerization-based bioprinting—process, materials, applications and regulatory challenges." Biofabrication 12, no. 2 (2020): 022001.

                                                   ii.     "Recent advances in formulating and processing biomaterial inks for vat polymerization‐based 3D printing." Advanced healthcare materials 9, no. 15 (2020): 2000156.

Author Response

                                                                                           09/November/2022

Dear Reviewer,

Thank you for sending the reviewer’s comments on our manuscript: “Expanding Quality by Design principles to support 3D printed medical device development following the renewed regulatory framework in Europe.

We are pleased that the reviewers find our review article fairly comprehensive covering a wide range of recent references on the topic. The reviewers requested some editing work and raised a list of valid points that significantly helped in improving the revised manuscript. All corrections appear in track changes and added sections are highlighted with yellow in the manuscript.

All authors have seen and agreed with the contents of the manuscript, and none has any financial interest. The submission is not under review elsewhere.

We hope that you now find the improved manuscript suitable for publication in Biomedicines and look forward to hearing from you.

Yours sincerely,

Livia Adalbert

Detailed response to reviewer

Reviewer #2:

We thank the reviewer for the constructive comments and suggestions. We feel that the manuscript has been considerably improved through the suggested changes.

Reviewer:

Overall, the paper is well-written. However, the authors need to make some minor changes such as using the standardized ASTM terms for different 3D printing techniques and elaborate more on bioprinting.

  1. In Section 1.1, the authors mentioned the different commonly used 3D printing techniques. The authors should use the ASTM classification for the 3D printing techniques and provide some relevant references for some of the mentioned techniques.
  2. Binder Jetting (binder jet printing)
  3. Directed-Energy Deposition
  4. Material Extrusion (fused deposition modelling, semi-solid extrusion)
  5. Material Jetting (inkjet printing)
  6. Powder-Bed Fusion (selective laser sintering)
  7. Sheet Lamination
  8. Vat polymerization (stereolithography)

 Author reply:

It has been corrected. ASTM classification for the 3D printing techniques with references have been provided.

1.1. What does 3D printing or AM mean?

33DP bridges art and science to print in a new dimension, applying 3D printers to metamorphose 3D computer-aided designs (CAD) into life-changing products, creating more effective and patient-friendly pharmaceutical products and bio-inspired medical devices [11]. 3DP was first developed almost three decades ago and it shook up the entire industrial and scientific fields, providing swift and precise manufacturing of structures and components with such high level of complexity that were not available via conventional methods [12]. It covers a broad range of techniques, such as deposition, binding, or polymerization of materials in successive layers for the manufacture of a variety of drug delivery systems, medical devices, and complex biomedical employments [13-15].

In 2010, the American Society for Testing and Materials (ASTM) laid down a set of standards which classifies the range of additive manufacturing processes into 7 categories based on the material and technology used [16]. Medical applications of additive manufacturing can be categorized as: 1) Medical models; 2) Implants; 3) Tools, instruments, and parts for medical devices; 4) Medical aids, supportive guides, splints, and prostheses; 5) Biomanufacturing [17]. The seven ASTM standardized AM technologies are:

  1. A) Binder jetting technique uses, two materials: a binder and a powder-based material. The building material is present in the powder form and the binder is in liquid form as it acts as an adhesive between the layers of powder. Binder jetting also allows color printing and uses polymers such as acrylonitrile butadiene styrene, polyamide, polycarbonate, metal (stainless steel) and ceramic materials (glass) in this process. This technique of printing is self-supported within the power bed. Typically, binder jetting is not used for tools, instruments, and parts for medical devices [18-21].
  2. B) Directed energy deposition includes a more complex printing process that is used to repair or add additional material to existing components. This process uses material in either powder or wire form. The machine consists of a nozzle placed on a multi axis arm through which melted material provided onto the surface and then it solidifies. This technology is like material extrusion process, but the nozzle moves into all the directions as it is not fixed to a specific axis. Typical thickness of layer falls between 0.25mm- 0.5mm. The process uses metals as materials such as cobalt chrome and titanium. This technique is utilized rarely only in implants [22, 23]
  3. C) Material Extrusion is the most common technique used for 3 D printers and a common extrusion process is fused deposition modelling. This technique prints layer by layer by drawing the material through nozzle, where it is heated [24, 25]. The nozzle moves horizontally, and the platform moves vertically up and down after each layer is formed. The typical thickness of layer should fall between 0.178mm-0.356 mm [26]. The material extrusion process is beneficial process as it can create readily available models with good structural and high-quality properties. This technique’s medical application is broad [20, 21, 27]
  4. D) Material jetting involves building of model or structure, layer by layer. Material is jetted through the nozzle on to the building platform either by using drop on demand technique or continuous approach[28, 29]. The material is filled in the nozzle which then moves horizontally on the building platform. Later, the layers can be cured by using UV light. The commonly used materials are polymers and waxes because of their viscous nature and ability to form drops. Material jetting is not utilized for implants and biomanufacturing. Polymers used are Polypropylene, HDPE (High density Polyethylene), PS (Polystyrene), PMMA (Polymethylmethacrylate), PC (Polycarbonate), ABS (Acrylonitrile Butadiene Styrene) & HIPS (High Impact Polystyrene) [17, 30]
  5. E) Power bed fusion either uses electron beam or laser to melt down and fuse material and powder together. It includes the commonly used techniques such as selective heat sintering, selective laser melting, selective laser sintering, electron beam melting and direct metal laser sintering [31]. All these processes include spreading of powder material over previous layers. This can be performed by different mechanisms, such as a roller or a blade. Fresh material is supplied by a hopper or reservoir. This technique’s medical application is broad [20, 21, 32]
  6. F) Sheet lamination includes 2 types of techniques: ultrasonic additive manufacturing and laminated object manufacturing. Ultrasonic additive manufacturing involves the use of metal sheets or ribbons such as aluminum, copper, stainless steel and titanium, which are then bound together by using ultrasonic welding. Laminated object manufacturing technique involves a layer-by-layer approach and uses paper as material and adhesive instead of welding. This technology is utilized rarely for medical models or phantoms [17, 20]
  7. G) VAT Photopolymerization uses a model that is created layer by layer using vat of liquid photopolymer resin [33]. An UV light is used to cure the resin into the desired model and the platform moves downwards accordingly to build new layers on the top of previous ones [34]. The typical thickness of layer for the process should be 0.025-0.5 mm. As this technique uses liquid for forming models or structures, there is no structural support provided during the printing phase [35]. Hence, in this case, support structures are needed to be added. After completion of the process, the model must be removed from the resin and the excess resin present in vat should be drained. This technique’s medical application is broad [20, 21, 31].

  1. In the 2nd part of Section 1.2, the term “bioprinting” should be used when cells are involved during the printing process. The authors can refer to some of the highly cited papers in the field of bioprinting.
  2. "Print me an organ! Why we are not there yet." Progress in Polymer Science97 (2019): 101145.
  3. "Bioinks and bioprinting technologies to make heterogeneous and biomimetic tissue constructs." Materials Today Bio1 (2019): 100008.

  1.  The authors should also discuss what are some of the bioprinting techniques that are suitable for cell printing? The authors can refer to some recent papers below.
  2. Material extrusion
  3. "Extrusion bioprinting of soft materials: An emerging technique for biological model fabrication." Applied Physics Reviews 6, no. 1 (2019): 011310.
  4. "Layer-by-layer ultraviolet assisted extrusion-based (UAE) bioprinting of hydrogel constructs with high aspect ratio for soft tissue engineering applications." PLoS One 14, no. 6 (2019): e0216776.
  5. Material jetting
  6. "Inkjet bioprinting of biomaterials." Chemical Reviews 120, no. 19 (2020): 10793-10833.
  7. "Controlling Droplet Impact Velocity and Droplet Volume: Key Factors to Achieving High Cell Viability in Sub-Nanoliter Droplet-based Bioprinting." International Journal of Bioprinting 8, no. 1 (2022). 424
  8. Vat polymerization
  9. "Vat polymerization-based bioprinting—process, materials, applications and regulatory challenges." Biofabrication 12, no. 2 (2020): 022001.
  10. "Recent advances in formulating and processing biomaterial inks for vat polymerization‐based 3D printing." Advanced healthcare materials 9, no. 15 (2020): 2000156.

Author reply:

3D bioprinting section has been more elaborated and highly cited papers have been included.

Additional section:

1.2 Bioprinting

Various type of additive manufacturing techniques have been developed which includes using cells and biomaterials for fabrication of tissues and organs such as  inkjet-based 3D bioprinting, laser assisted 3D bioprinting, stereolithographic based 3D bioprinting and extrusion-based 3D bioprinting [86].

 Inkjet based 3D bioprinting involves passing of sequential drops of bioinks on a surface using thermal piezoelectric or electromagnetic effect. It includes usage of cells and biomaterials. Cells include neonatal human dermal fibroblasts, dermal microvascular endothelial cells, epidermal keratinocytes and human chondrocytes. Biomaterials include collagen, fibrinogen, thrombin, poly (ethylene glycol) & dimethacrylate (PEGDMA). This technique can be used for better wound contraction and tissue integration and comes with some benefits such as it is easily available, cheap in cost and is a high-speed technique but it also has lack of precision [87-89].

Extrusion-based 3D bioprinting involves passing of bioink through nozzle under pneumatic or mechanical forces. Cells used as human keratinocytes and fibroblasts, chondrocytes, HUVECs (Human umbilical vein endothelial cells) and MSCs (mesenchymal stem cells). Biomaterials include alginate, hyaluronic acid, fibrinogen, glycerol, collagen, and gelatin. This technique is be used for reconstruction of facial wounds, cartilage or 3DP of endothelialized-myocardium-on-a chip. The benefit of this technique is that high viscosity bioink can be used and cells can be printed in greater density but cell’s structure can be distorted because of the undue stress during density [24, 90].

Laser assisted 3D bioprinting is used to evaporate bioink and cell suspension placed at the bottom of ribbon, which is then moved on to the receiving substrate. The following cells are used such as HUVECs, mouse fibroblasts, human keratinocytes and MSCs. Biomaterials cover collagen, nanohydroxyapatite and human osseous cell sheets. This technique can be used for soft tissue regeneration and bone and skin grafting. High degree of precision and resolution can be achieved in this process, but it is expensive and time-consuming technique [91-93].

Stereolithography based 3D bioprinting is used to cure photocurable bioink in a layered manner. Cells include MCF-7 breast cancer cell, HUVECs, C2C12 skeletal muscle cells, osteoblasts, fibroblasts, mesenchymal cells, BrCa (breast cancer) and MSCs. Biomaterials cover: PEGDA (Poly (ethylene glycol) diacrylate) and GeIMA, GeIMA and nHA (Nano-hydroxyapatite). This technique can be used for preparing a model for post-metastatic breast cancer progression investigation in bone. High degree of accuracy can be achieved with low printing time, but this technique is highly complicated and also has lengthy post-processing process [94-96].

Generally, native tissues are of more complex structures than engineered constructs, therefore, it is critical to develop functional and biomimetic tissue-like constructs [97]. It is also important to consider the different development stages that the printed tissues often need to undergo, including cell viability, implantation, integration, and remodeling in vivo. Mimicking heterogeneous and complex native tissues have been made possible by bioprinting, by using multicomponent crosslinkable bioinks that can solidify to maintain stable constructs. In principal, shear-thinning biomaterials, that are injectable under application of shear force, show capacity to quickly self-heal, are popular as they reform once the shear stress is removed [98]. Challenges related to multicomponent bioinks include fast degradations properties of hydrogels that have been used for 3D bioprinting of heterogenous and biomimetic structures.

Round 2

Reviewer 1 Report

The revisions made are perfect.